# Practical considerations for Ultraviolet-C radiation mediated decontamination of N95 respirator against SARS-CoV-2 virus

Guillaume R. Golovkine[1][◉], Allison W. Roberts[1][◉], Chase Cooper[2], Sebastian Riano[2], Angela M. DiCiccio[2], Daniel L. Worthington[2], Jeffrey P. Clarkson[2], Michael Krames[3], Jianping Zhang[4], Ying Gao[4], Ling Zhou[4], Scott B. Biering[5], Sarah A. Stanley[1,5]*

1 Department of Molecular and Cell Biology, University of California, Berkeley, California, United States of America, 2 Verily Life Sciences, South San Francisco, California, United States of America, 3 Arkesso, LLC, Palo Alto, California, United States of America, 4 Bolb Inc, Livermore, California, United States of America, 5 School of Public Health, University of California, Berkeley, California, United States of America

◉ These authors contributed equally to this work.
* sarah.stanley@berkeley.edu

**Data Availability Statement:** All relevant data are within the manuscript and its Supporting Information files.

## Abstract

Decontaminating N95 respirators for reuse could mitigate shortages during the COVID-19 pandemic. Although the United States Center for Disease Control has identified Ultraviolet-C irradiation as one of the most promising methods for N95 decontamination, very few studies have evaluated the efficacy of Ultraviolet-C for SARS-CoV-2 inactivation. In addition, most decontamination studies are performed using mask coupons that do not recapitulate the complexity of whole masks. We sought to directly evaluate the efficacy of Ultraviolet-C mediated inactivation of SARS-CoV-2 on N95 respirators. To that end we created a portable UV-C light-emitting diode disinfection chamber and tested decontamination of SARS-CoV-2 at different sites on two models of N95 respirator. We found that decontamination efficacy depends on mask model, material and location of the contamination on the mask. Our results emphasize the need for caution when interpreting efficacy data of UV-C decontamination methods.

## Introduction

The limited availability of N95 respirators during the SARS-CoV-2 pandemic has forced many healthcare workers to reuse respirators designed for one-time use. In these circumstances, the development of safe and efficient methods of decontamination of N95 respirators could be a partial solution to shortages [1]. The U. S. Center for Disease Control (CDC) has identified ultraviolet germicidal irradiation (UVGI), vaporous hydrogen peroxide and moist heat as the 3 most promising methods for N95 decontamination during a crisis capacity situation [1, 2].

The efficacy of UVGI for decontamination of bacteria and viruses on N95 respirators has been extensively investigated [3]. Ultraviolet-C (UV-C) exposure has been identified as an efficient method for inactivation of several viruses [4, 5], including respiratory viruses such as influenza [6–8], SARS-CoV or MERS-CoV [9]. However, important variability was reported

**Funding:** This work was funded by Fast Grants (part of Emergent Ventures at George Mason University) to SAS. AWR is an Open Philanthropy Fellow of the Life Sciences Research Foundation. Authors CC, SR, AMD, DLW and JPC are employees of Verily Life Sciences. Authors JZ, YG, LZ are employees of Bolb, Inc. Author MK is President at Arkesso, LLC. The funders provided support in the form of salaries for authors CC, SR, AMD, DLW, JPC, JZ, YG, LZ, MK and support in the form of work hours and equipment for the authors to execute and collaborate in the study design, data collection and analysis. The funders permitted authors to proceed with the decision to publish, or preparation of the manuscript.

**Competing interests:** Authors CC, SR, AMD, DLW and JPC are employees of Verily Life Sciences, the company that designed and built the UV-C device. Authors JZ, YG, LZ are employees of Bolb, Inc, the company that created the LEDs used in the study. Author MK is President at Arkesso LLC, a contracted advisor to Bolb, Inc. Verily Life Sciences and Arkesso, LLC do not claim any patents or commercial products related to the study subject. Bolb, Inc. does not claim any patents on the study subject but does have the following patents covering the SMD6060 LEDs used in the study: - Lattice-constant formatted epitaxial template for light emitting devices and a method for making the same, US9,472,716. - Light emitter with a conductive transparent p-type layer structure, US9,293,648. - Light emitter with a conductive transparent p-type layer structure, US9,553,232. - Ultraviolet light-emitting device with a heavily doped strain-management interlayer, US9,680,056. - Ultraviolet light device, US9, 715,058. This does not alter our adherence to all the PLOS ONE policies on sharing data and materials.

depending on the N95 mask model studied [6, 9]. Very few studies directly evaluate UV-C mediated inactivation of SARS-CoV-2 on N95 respirators [10–12]. Fischer et al. demonstrated that UV-C could effectively decontaminate SARS-CoV-2 on N95 respirator [11]. However, this study was performed using small, flat mask coupons that do not recapitulate angular incidence and shadowing effects caused by the 3D structure of the masks and could therefore underestimate the levels of UV-C irradiation required for effective decontamination of an intact respirator [13]. Recently, Ozog et al. tested UV-C decontamination on whole N95 respirators and reported important variations of decontamination efficacy between different N95 models and material [12], which correlates with previous results with H1N1 influenza [6, 9]. However, in this study the virus recovery from unirradiated masks for several of the models was not sufficiently above the limit of detection to determine whether effective decontamination was achieved as defined the U.S. Food & Drug Administration (FDA) as a minimum of 3 $\log_{10}$ reduction in viable virus [14].

The COVID-19 pandemic and the associated shortage in N95 supplies have triggered the rapid emergence of new implementation strategies for decontamination methods and the creation of new UVGI devices [2]. During the first months of the COVID-19 pandemic, the University of Nebraska Medical Center, followed by other groups, published protocols for the implementation of UV-C based decontamination of N95 [15]. In April 2020, during the peak on the COVID-19 pandemic, the Henry Ford Health System and other hospital settings used UV-C to decontaminate N95 respirators for health care workers [12, 16]. However, as of April 2021, only one UVGI device had received an Emergency Use Authorization (EUA) from the FDA [14].

We aimed to determine the efficacy of UVGI for decontamination of SARS-CoV-2 on intact N95 respirators. To evaluate whether the 3D structure of the masks impacted inactivation of SARS-CoV-2, we tested decontamination at several sites on the respirators. We found that the efficacy of decontamination is significantly influenced by the structure of the mask and corresponding differences in irradiation. We also sought to directly evaluate whether the efficacy of decontamination varies between different models of N95 made with different materials.

# Materials and methods

## Virus preparation

The SARS-CoV-2 strain used was USA-WA1/2020. Viral stocks were obtained from the Biodefense and Emerging Infections Research Resources Repository. Stocks were amplified in Vero-E6 cells (passage 1) and again in Calu-3 cells (passage 2). Virus passage 2 was used for experiments and was determined to have a concentration of $8 \times 10^7$ TCID$_{50}$/ml. Additional details in S1 File.

## Mask inoculation

Five locations (center, top, bottom, right cheek, and strap) on the exterior of the masks were inoculated with a total of 50 μl of virus stock. The locations were selected after consideration of the 3D structure of the mask and corresponding differences in irradiation dose received to cover a wide range of anticipated doses. Aluminum coupons adhered to the center of the masks were used as a smooth and non-porous control surface. Aluminum coupons were bent to follow the shape of the mask and placed in a location calculated to receive irradiation doses equivalent to the "center" mask location.

To allow surface tension to contain the virus in droplets on non-horizontal surfaces, virus was inoculated in 3 aliquots of 16.67 μl. The 3 aliquots were inoculated simultaneously and

spaced such that they could not merge but all fit within the size of a 12 mm biopsy punch. Masks were left to dry for 3.5 hours in a biosafety cabinet (Nuaire LabGard model NU-540-600). Straps of 3M 1860 masks were inoculated with 50 μl of SARS-CoV-2 only 10 minutes before irradiation because optimization experiments showed that virus viability on this material decreased with excessive drying (unpublished data). Desiccation on mask facepieces or aluminum coupons for 3.5 hours did not significantly affect virus concentration (S3 Fig).

## UV-C exposure

We created a UVGI device for N95 decontamination designed to generate high levels of reflection and enable ease of use via straightforward fixturing and application. The decontamination chamber consists of a metal reflecting box containing high power, commercially available UV LEDs with driver circuitry on metal core printed PCBs mounted on the vertical walls of the chamber (S1 Fig). Individual masks were placed inside the chamber by attaching their upper and lower head bands to attachment points that ensured consistent placement inside the chamber. Eight LEDs were arrayed on each vertical sidewall in a fashion to optimize exposure dose uniformity across the surface of an N95 respirator and were calibrated to deliver a minimum irradiance of 1 mW/cm$^2$ across all locations of the mask (S2 Fig and S1 Table). The temperature of the metal core circuit board, to which the UV LEDs were mounted to, remained below 41˚C for all disinfection runs.

Immediately before irradiation, 2 pieces of UV tape were added to the upper right and lower left corners of the mask. Masks were placed one at a time into the device by attaching the straps on mounting points located at the top and bottom. Masks were irradiated for 300 sec or 600 sec. A picture of the mask was taken after irradiation to document UV tape change of color. To ensure that no loss of virus viability occurred due to desiccation time, virus from non-exposed control masks were harvested after all masks were exposed and biopsies were taken. Control masks were not placed into the device.

## Virus titration

Inoculated regions of the mask were cut out using 12 mm biopsy punches. Samples were incubated in 1.4 ml (mask punches and aluminum coupons) or 2 ml (strap pieces) of DMEM (Sigma-Aldrich) supplemented with 10% fetal bovine serum, 100 U/mL penicillin, and 100 μg/mL streptomycin for a minimum of 30 minutes and subjected to gentle manual agitation at the beginning and end of the incubation. Virus was quantified by TCID$_{50}$ assay by incubating Vero E6 cells in 96 well plates with 10-fold serial dilutions in 8-fold of incubation media. Five days after inoculation, cytopathic effect, defined as any virus induced cell death or change in cell morphology, was scored visually under brightfield illumination using a 4X / 0.13 NA objective. TCID$_{50}$ was calculated using the Reed-Muench method. The limit of detection of the assay was 3.16 TCID$_{50}$/ml, which was determined by calculating the TCID$_{50}$ at which no CPE is observed in any replicate wells. The maximum log reduction that could be achieved for each mask location is shown in S2 Table.

## Results and discussion

We tested decontamination of SARS-CoV-2 on two masks models, 3M 1860 and 3M 8210. Although both masks are approved for healthcare worker use during the COVID-19 pandemic, their outer layers have different shapes and are comprised of different materials. We wondered whether these differences would impact the efficacy of UV-C based decontamination. We analyzed decontamination of 5 different inoculated mask locations (center, top, bottom, right cheek and strap, Fig 1 panel A) as well as a control aluminum coupon adhered to

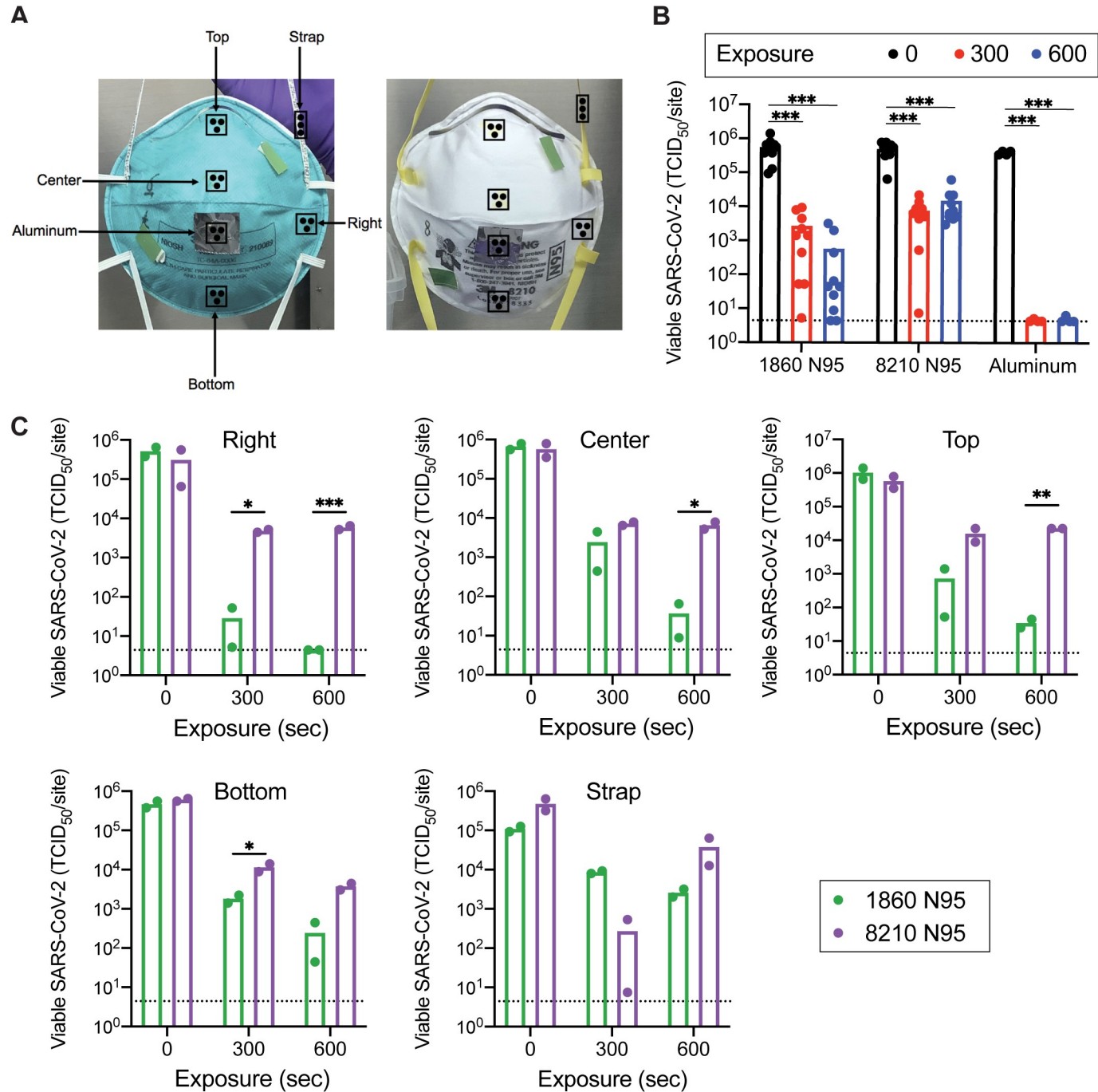

**Fig 1. UV-C decontamination of multiple locations on two models of N95 respirators.** (A) Schematic of mask inoculation sites. 3M 1860 and 3M 8210 masks were inoculated in five different locations plus an aluminum coupon adhered to the center of each mask. Each site was inoculated with 50 µl of $8e^7$ $TCID_{50}$/ml virus, applied as three aliquots of 16.7 µl. Inoculated masks were allowed to dry for 3.5 hours at room temperature in a biosafety cabinet before masks were exposed to UV-C irradiation. UV tape was adhered to masks to confirm irradiation. (B and C) Viable SARS-COV-2 recovered from inoculation sites. Viable virus at each inoculation site was quantified by end-point titration on Vero E6 cells and expressed as 50% tissue-culture infectious dose 50 ($TCID_{50}$) per site. Plots show the mean of two replicates from one experiment and are representative of two independent experiments. Dashed lines indicate the limit of detection (LOD), samples with no positive wells are plotted at LOD. (B) Data displayed by mask model; mean of data is displayed as a bar graph with individual sites shown as dots. (C) Data displayed by location of inoculation. * $p < 0.05$, ** $p < 0.01$, *** $p < 0.001$.

center of the mask. Masks were exposed to UV-C for 0, 300 or 600 seconds. The minimal doses received at each location were greater than 300 and 600 mJ/$cm^2$ for the 300 and 600 second exposures, respectively. Importantly, the UV-C doses used in our study were considerably lower than the cumulative doses reported to degrade N95 material [17].

The CDC has issued specific recommendations on the reuse of N95 respirators as a crisis management strategy. Accordingly, the FDA requires "Tier 3" devices (bioburden reduction) to yield at least 3 $\log_{10}$ inactivation of various pathogens on N95 respirators [14]. Both UV-C doses achieved close to a 5 $\log_{10}$ reduction in virus on the aluminum control coupons (Fig 1, panel B and Table 1), validating the efficacy of our UV-C device to eliminate SARS-CoV-2 on non-porous material. However, the 300 second exposure was insufficient for decontamination when averaging locations across the masks (Fig 1, panel B and Table 1). The 600 second exposure effectively decontaminated the 3M 1860 masks but failed to decontaminate 3M 8210 masks (Fig 1, panel B and Table 1). Notably, there was little difference between the 300 and 600 second doses on the 8210 masks regardless of location, suggesting that increased exposure time does not achieve higher levels of decontamination of this mask surface.

While the reduction averaged across the entire mask was greater than 3 $\log_{10}$ for the 3M 1860 mask at 600 seconds, there were important variations between different locations on the mask (Fig 1, panel C and Table 1). Irradiation doses were calculated using a representative 1860 model N95 mask with integrated irradiance sensors (S2 Fig). We determined that the smaller reduction in viral titer at the bottom location correlates with a lower irradiation dose received at this location. Conversely, the Right location received the highest irradiation dose and demonstrated effective decontamination of the 3M 1860 masks at the 300 sec and 600 sec exposures. However, neither dose at the Right location was sufficient for decontamination of the 3M 8210 masks (Fig 1, panel C), despite receiving the largest dose. Furthermore, the straps were difficult to decontaminate, with large variability (Fig 1, panel C), a result observed for other viruses [8]. We hypothesize that this is due to the strap material and potential shadowing effects caused by twists in the strap during exposure to UV-C.

These results suggest that some N95 respirator models are not compatible with UV-C based decontamination. Importantly, dose validation experiments were performed with both mask models and showed similar irradiances at each location. Therefore, differences in decontamination efficacy do not result from variations in the 3D shape of the masks but are likely due to differences in mask material. The 1860 model is designed to be fluid resistant and has a smooth polypropylene outer layer while the 8210 model is not considered fluid resistant and has a polyester outer layer. Our results suggest that the 1860 facepiece is more appropriate for UV-C decontamination than the 1860 strap (braided polyisoprene), the 8210 facepiece, or the 8210

**Table 1. Average $\log_{10}$ reduction in viable SARS-CoV-2 recovered.**

|  | 300 sec | | 600 sec | |
|---|---|---|---|---|
|  | **3M 1860** | **3M 8210** | **3M 1860** | **3M 8210** |
| **Total mask** | 2.67 | 2.05 | 3.74 | 1.68 |
| **Aluminum** | 4.95 [4.95–4.95] | 4.92 [4.88–4.95] | 4.95 [4.95–4.95] | 4.86 [4.79–4.95] |
| **Right** | 4.25 [3.99–4.99] | 1.81 [1.78–1.85] | 5.06 [5.06–5.06] | 1.73 [1.68–1.78] |
| **Center** | 2.44 [2.18–3.18] | 1.90 [1.86–1.94] | 4.26 [4.01–4.88] | 1.94 [1.86–2.04] |
| **Top** | 3.15 [2.86–4.29] | 1.56 [1.41–1.81] | 4.47 [4.36–4.61] | 1.41 [1.41–1.41] |
| **Bottom** | 2.41 [2.32–2.52] | 1.72 [1.63–1.83] | 3.28 [3.02–4.02] | 2.21 [2.13–2.30] |
| **Strap** | 1.10 [1.07–1.14] | 3.24 [2.95–4.80] | 1.63 [1.54–1.74] | 1.10 [0.88–1.58] |

Data is presented as the mean with lowest and highest values within brackets.

straps (thermoplastic elastomers). Both the 1860 and 8210 facepieces are hydrophobic and virus inoculum was not absorbed into the material. Instead, virus droplets dried on the mask surface. Although, no loss of viability was observed during the 3.5 hours of drying time (S3 Fig), it is possible that desiccation could increase susceptibility of the virus to UV-C. Interestingly, we noted that the outer layer of the 8210 mask presents a rougher surface than the 1860 mask, which perhaps shields the virus from sufficient UV-C exposure and prevents efficient UV-C decontamination of this mask model.

Although our device was designed to expose the entirety of the masks to UV-C, our study focused on the decontamination of the outer layer of the mask only, which limits the conclusions to single-user applications. Although a system shown to effectively decontaminate both the interior and exterior of masks could streamline decontamination and allow for multiple-users, the use of respirators by single-users poses less risk of unintended transmission of SARS-CoV-2 or other pathogens [14] and could represent a viable crisis management strategy to help alleviate N95 shortages. It is important to note that UV-C decontamination methods should also ensure that N95 masks retain their fit and filtration capacities after UV-C exposure [18], which was not tested in this study.

While UV-C is an attractive method for decontamination of PPE when applied at appropriate doses that do not compromise material integrity and device functionality, our findings suggest that efficacy for individual mask models should be evaluated for a given UV-C device. Our results as well as the recent study by Ozog et al. [12] indicate that while the facepieces of some mask models can be successfully decontaminated using UV-C, others seem incompatible with this method of SARS-CoV-2 decontamination. Important factors to consider are the 3D structure of the mask and corresponding differences in irradiation dose received in some mask locations which can significantly influence the efficacy of decontamination. The straps may be particularly difficult to decontaminate and may require the use of a secondary method of decontamination in addition to UVGI [19]. However, UV-C LED technology is improving rapidly, and future devices will offer higher irradiation levels, improving penetration of UVGI and/or shortening exposure times. The identification of existing N95 models that are most suited for UV-C based decontamination or the creation of new mask models for this purpose would be important milestones that could help mitigate future N95 shortages.

## Supporting information

**S1 File. Extended material and methods.**
(DOCX)

**S2 File. Minimal data set.**
(XLSX)

**S1 Fig. Photo of a N95 respirator in the UVGI device.** Front panel was removed for visibility.
(TIF)

**S2 Fig. Custom N95 respirator with calibrated sensors used for the measurement and according irradiance measurement at each site.**
(TIFF)

**S3 Fig. Comparison of virus recovery from unirradiated mask sites after virus desiccation to control virus added directly into recovery media.**
(TIFF)

**S1 Table. Calculated doses delivered at each location for each exposure.** Doses are calculated based on irradiance measurements made with the custom N95 respirator with calibrated

sensors. Units are in mJ/cm$^2$.
(DOCX)

**S2 Table. Maximum log reduction achievable for each mask location.**
(DOCX)

## Acknowledgments

We thank Verily employees Greg Arcenio, Beth Bosworth, Warren Cai, Mike Chen, Junjia Ding, Tim English, Chopin Hua, David Heinz, Kyle Nichols, Supriyo Sinha for their valuable contributions and feedback.

## Author Contributions

**Conceptualization:** Allison W. Roberts, Angela M. DiCiccio, Daniel L. Worthington, Jeffrey P. Clarkson, Sarah A. Stanley.

**Data curation:** Guillaume R. Golovkine, Allison W. Roberts.

**Formal analysis:** Guillaume R. Golovkine, Allison W. Roberts.

**Funding acquisition:** Sarah A. Stanley.

**Investigation:** Guillaume R. Golovkine, Allison W. Roberts, Scott B. Biering.

**Methodology:** Guillaume R. Golovkine, Allison W. Roberts, Chase Cooper, Sebastian Riano, Angela M. DiCiccio, Daniel L. Worthington, Jeffrey P. Clarkson, Scott B. Biering, Sarah A. Stanley.

**Project administration:** Sarah A. Stanley.

**Resources:** Chase Cooper, Sebastian Riano, Angela M. DiCiccio, Daniel L. Worthington, Jeffrey P. Clarkson, Michael Krames, Jianping Zhang, Ying Gao, Ling Zhou, Sarah A. Stanley.

**Validation:** Guillaume R. Golovkine, Allison W. Roberts.

**Visualization:** Allison W. Roberts.

**Writing – original draft:** Guillaume R. Golovkine, Allison W. Roberts.

**Writing – review & editing:** Guillaume R. Golovkine, Allison W. Roberts, Angela M. DiCiccio, Daniel L. Worthington, Jeffrey P. Clarkson, Sarah A. Stanley.

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
