## [Decision Letter · Decision Letter 0]

28 Apr 2021

PONE-D-21-00512

Practical considerations for Ultraviolet-C radiation mediated decontamination of N95 respirator against SARS-CoV-2 virus

PLOS ONE

Dear Dr. Stanley,

Thank you for submitting your manuscript to PLOS ONE. After careful consideration, we feel that it has merit but does not fully meet PLOS ONE’s publication criteria as it currently stands. Therefore, we invite you to submit a revised version of the manuscript that addresses the points raised during the review process.

I am returning your manuscript with comments from two reviewers, who you will see, came to similar conclusions. Both found the paper very interesting but had a number of queries about the methods used. I agree that the methods section is currently very brief and important information is either missing or included as supplementary material rather than main text. For example, the treatment of control (non-exposed) masks/surfaces should be clarified - was virus recovered after the 3.5 h drying time (0 sec exposure) or were masks placed in the chamber and exposed to 0 mJ/cm2 for 300 and 600 seconds? Similarly, straps were not subjected to the 3.5 h drying time and, whilst it is stated (supplementary data) that no losses in viability occurred over the 3.5h drying time, could desiccation stress increase virus susceptibility to subsequent UV-c exposure? Whilst some of the additional information asked for by the reviewers can be considered supplementary, please ensure sufficient detail is provided within the main text. If possible, and as suggested by both reviewers, please also include a picture of the chamber and configuration of the masks within.

I would also draw specific attention to comments made by Reviewer 2 and their concerns relating to data interpretation. Please ensure your conclusions are supported by the data and that limitations of the study have been acknowledged.

Finally, and as highlighted by Reviewer 1, it could be perceived that some of the listed authors have competing interests. Please address and/or declare as appropriate.

We look forward to receiving your revised manuscript.

Kind regards,

Ginny Moore

Academic Editor

PLOS ONE

Journal Requirements:

[This work was funded by Fast Grants (part of Emergent Ventures at George Mason University) to SAS. AWR is supported by a LSRF fellowship.].   

We note that one or more of the authors are employed by a commercial company: Verily Life Sciences, Arkesso LLC and Bolb Inc.

Reviewers' comments:

Reviewer's Responses to Questions

**Comments to the Author**

1. Is the manuscript technically sound, and do the data support the conclusions?

Reviewer #1: Yes

Reviewer #2: Partly

2. Has the statistical analysis been performed appropriately and rigorously? 

Reviewer #1: I Don't Know

Reviewer #2: Yes

3. Have the authors made all data underlying the findings in their manuscript fully available?

Reviewer #1: Yes

Reviewer #2: Yes

4. Is the manuscript presented in an intelligible fashion and written in standard English?

Reviewer #1: Yes

Reviewer #2: Yes

5. Review Comments to the Author

Reviewer #1: Review of PONE-D-21-00512

Interesting short report showing differences in mask decontamination. Could do with more results because it was only completed twice for each mask and the results lack some potential significance in interpretation due to it. It would also be interesting to have completed the study on masks that had already been worn because this is how they would be presented when decontaminated in reality. There is no evaluation of the effects angular incidence causes on the decontamination results as mentioned in the introduction.

Conflicts of interest, there are authors from the companies that have produced the LEDs for the UV chamber you have used. I would say that this needs to be declared since you say that UVc is a technology that can be used to disinfect masks and will be improving in the future which could benefit the companies producing the LEDs.

Why did you choose the sites for the contamination of the masks?

Why did you only test two models of the mask? Was it because these were the only ones being used in the University or were they chosen because their properties were thought to give different results from the exposure?

The last sentence in the introduction is not needed.

How did you stop the droplet inoculum from spreading on the non-horizontal surfaces if they were all inoculated at the same time? How does this vary from just adding 50ul in one go? Were the 3 inoculums added to exactly the same spot? Did you add each inoculum and leave it to dry prior to adding the next? If not added together how does this represent contamination when masks are used in practice?

More information on the UVc exposure chamber is needed and positioning of the masks inside it, or was each mask exposed separately? Maybe a picture of the chamber would help in the description, the brief section in the results/discussion section should be moved into the materials and methods. Was there a temperature increase in the exposure device during operation?

More information on the methods is needed or should be included in the supplementary file. How was the CPE visualised, was a stain used?

Please explain what were the aluminium coupons used for (virus titration section)? Why were aluminium coupons chosen over stainless steel for coupons?

Were the coupons excised from the mask agitated in the recovery medium?

What was the loss of viability over the course of the experiment from exposure to the environmental conditions inside the UVc chamber without the LEDs being switched on? Could an increase in heat have contributed to the reduction in viability of the virus?

Please include a reference for the statement identifying that a 3-log reduction is the industry standard for an effective decontamination.

The results from the aluminium show a close to 5 log reduction, not an actual 5 log reduction. The text needs to be changed to reflect this. How were the aluminium coupons presented in the chamber and how does this relate to the inoculated masks (distance, orientation)? What was the maximum log reduction that could be achieved in this assay?

Some more interpretation of the surface types would be good for the discussion. You mention the 1860 was hydrophobic, but being made from polyester, was the 8210 hydrophobic as well? How did the inoculum present on the surfaces i.e. did they stay as droplets on the material or were they absorbed, and how do you think this would have affected the exposure? Do you think the colour differences of the masks played a role in the decontamination?

How close were your exposure doses to those that would compromise material integrity?

Supplementary file

You mention that “Details on virus propagation can be found in supplementary materials” but that is in the supplementary file. Please clarify.

What type of Biosafety cabinet was used for the drying of the masks?

Reviewer #2: The paper entitled “Practical considerations for Ultraviolet-C radiation mediated decontamination of N95 respirator against SARS-CoV-2 virus” is well written however would benefit from a more comprehensive review of the literature in the introduction and discussion . The contents add to the body of literature on decontamination of respirators and discuss important information on practical considerations for these processes. Of great importance is their discussion on variability in efficacy of UV-C depending on respirator model and mode of testing (swatches vs. whole masks). The article could be improved upon by including a more in depth look at current respirator UV-C disinfection practices and the advantages / disadvantages of these methods. I have a few minor comments and questions.

Questions and comments:

Page 2. In the introduction, address if any facilities are using UV-C decontamination for respirators currently or during the peak of the pandemic? This would put the article into context with current practices and would be helpful either in the introduction or discussion. Examples are Ozog et al. 2020 and Golladay et al. 2021.

Page 2: In the introduction or discussion include the benefits / disadvantages to respirator disinfection with UV-C vs. other methods. This will give the reader a more comprehensive view on what is practical for their facility.

Page 3. Instead of referring to the ambiguous industry standard it would be better to cite the FDA’s “recommendations for sponsors requesting EUA’s for decontamination and bioburden reduction systems for surgical masks and respirators during the (COVID-19) public health emergency” document. Include in the discussion or introduction that there is currently only 1 UV-C device with an EUA for respirator disinfection.

Page 4. Mask inoculation and UV-C exposure – indicate that only the exterior of the mask was tested. This limits use of the respirator to the to the same user. Do you have a comment on importance (or lack there of?) of disinfecting the internal fibers and interior of the respirator?

Page 4. The description of the UV-C disinfection device should be under materials and methods, not results and discussion heading. If possible, it would be helpful to include an image of the device and placement of the mask in the supplementary files.

S1 File: Extended Material and Methods page 2 – indicate source and model for calibrated UV-C sensors.

Page 5. Figure: UV-C decontamination of multiple locations on two models of N95 respirators: Panel B) Indicate in the legend that it is showing each site independently with the dots then an average of all sites with the bar graph.

Page 5. Reference FDA recommendations instead of “industry standard”.

Page 7 line 3 – Were irradiation doses carried out on the 8210 model as well? If the irradiation doses on the 8210 and 1860 are the same in the various locations, then the material of the mask is the major influencing factor. It is possible that there is a small change in the shape of the 8210 compared to the 1860 that hinders the efficacy of UV-C.

Page 7 line 14 and 15 – if you conducted this same testing on swatches of mask would you see this same result? This would solidify that it is the material that is the major factor in efficacy.

Did treated masks pass fit and filtration testing? If not tested, it needs to be indicated as a limitation of the study.

Testing of only the exterior of the mask needs to be included as a limitation of the study.

Discussion: It would make sense that the center of the mask would have the highest irradiation levels since it is closes and most parallel to the LEDs however it does not. The best reduction is occurring at the edges of the mask, any thoughts on why that is?

6. PLOS authors have the option to publish the peer review history of their article (what does this mean?). If published, this will include your full peer review and any attached files.

Reviewer #1: No

Reviewer #2: No

---

## [Author Response · Author response to Decision Letter 0]

16 Jul 2021

We thank the reviewers for their feedback on the original manuscript and for the suggestions they made to improve our article. We have incorporated most of the suggestions made by the reviewers. A point-by-point response to the reviewers’ comments and questions can be found in the document named "Response to reviewers".

---

## [Decision Letter · Decision Letter 1]

27 Sep 2021

Practical considerations for Ultraviolet-C radiation mediated decontamination of N95 respirator against SARS-CoV-2 virus

PONE-D-21-00512R1

Dear Dr. Stanley,

We’re pleased to inform you that your manuscript has been judged scientifically suitable for publication and will be formally accepted for publication once it meets all outstanding technical requirements.

Kind regards,

Ginny Moore

Academic Editor

PLOS ONE

Additional Editor Comments (optional):

Please check the last line on page 2 of the revised document "4 of the 3 most promising methods"

Please check title of Table and confirm that "average" is the mean (or not)

Page 10 of the revised document - you mention "large variability". Whilst this is  illustrated to some extent in the figure, you could consider adding this data to the table (e.g. providing a standard deviation or range of reductions) 

Reviewers' comments:

Reviewer's Responses to Questions

**Comments to the Author**

1. If the authors have adequately addressed your comments raised in a previous round of review and you feel that this manuscript is now acceptable for publication, you may indicate that here to bypass the “Comments to the Author” section, enter your conflict of interest statement in the “Confidential to Editor” section, and submit your "Accept" recommendation.

Reviewer #1: All comments have been addressed

2. Is the manuscript technically sound, and do the data support the conclusions?

Reviewer #1: Yes

3. Has the statistical analysis been performed appropriately and rigorously? 

Reviewer #1: I Don't Know

4. Have the authors made all data underlying the findings in their manuscript fully available?

Reviewer #1: Yes

5. Is the manuscript presented in an intelligible fashion and written in standard English?

Reviewer #1: Yes

6. Review Comments to the Author

Reviewer #1: (No Response)

7. PLOS authors have the option to publish the peer review history of their article (what does this mean?). If published, this will include your full peer review and any attached files.

Reviewer #1: No

---

## [Editor Report · Acceptance letter]

1 Oct 2021

PONE-D-21-00512R1 

Practical considerations for Ultraviolet-C radiation mediated decontamination of N95 respirator against SARS-CoV-2 virus 

Dear Dr. Stanley:

I'm pleased to inform you that your manuscript has been deemed suitable for publication in PLOS ONE. Congratulations! Your manuscript is now with our production department. 

Kind regards, 

on behalf of

Dr. Ginny Moore 

Academic Editor

PLOS ONE